

Technical Note: Analytical Solution for Well Water Response to Earth Tides in
Leaky Aquifers with Storage and Compressibility in the Aquitard
Rémi Valois[1, 2, *], Agnès Rivière[3], Jean-Michel Vouillamoz[4], Gabriel C. Rau[5]
[1]French Red-Cross, 4 rue Diderot, Paris, France
[2]Hydrogeology Lab, UMR EMMAH, University of Avignon, 74 rue Louis Pasteur, Avignon, France
[3]Geosciences Department, Mines Paris - PSL, 75272 Paris, France
[4]Univ. Grenoble Alpes, IRD, CNRS, INRAE, Grenoble INP, IGE, 38000 Grenoble, France
[5]School of Environmental and Life Sciences, The University of Newcastle, Callaghan, Australia
*Correspondence: remi.valois1@gmail.com; Tel.: +33-6-8434-0779 (R.V.)

## Key points

- Development of a new analytical solution for Earth tide induced well water level fluctuations in
semi-confined aquifers considering aquitard storage, aquitard response to tidal strain, skin and
wellbore storage effects
- The solution correctly reflects previously observed but unexplained amplitude-frequency
relationships and positive or negative phase shifts
- Diagnostic information about subsurface hydro-geomechanical properties can be derived from
amplitude ratio and phase shifts for both semi-diurnal and diurnal tides

## Abstract

In recent years, there has been a growing interest in utilizing the groundwater response to Earth tides
as a means to estimate subsurface properties. However, existing analytical models have been
insufficient in accurately capturing realistic physical conditions. This study presents a new analytical
solution to calculate groundwater response to Earth tide strains, including storage and compressibility
of the aquitard, borehole storage and skin effects. We investigate the effects of aquifer and aquitard
parameters on well water response to Earth tides at two dominant frequencies ($O_1$ and $M_2$) and
compare our results with hydraulic parameters obtained from a pumping test. Inversion of the six
hydro-geomechanical parameters from amplitude response and phase shift of both semi-diurnal and
diurnal tides provides relevant information about aquifer transmissivity, storativity, well skin effect,
aquitard hydraulic conductivity and diffusivity. The new model is able to reproduce previously
unexplained observations of the amplitude and frequency responses. We emphasize the usefulness in
developing relevant methodology to use the groundwater response to natural drivers for
characterizing hydrogeological systems.



## 1. Introduction

Aquifer properties play a vital role in managing groundwater resources, particularly amid increasing anthropogenic groundwater use and the impact of climate change. While pump testing can be costly, there exists a cost-effective alternative for assessing aquifer hydraulic properties - analysing the groundwater response to Earth tides or atmospheric tides (McMillan et al., 2019). Observations of variations in groundwater level due to tidal fluctuations date back to the works of Klönne (1880), Meinzer (1939), and Young (1913). However, it was only later that hydro-geomechanical models were employed to elucidate these variations (Bredehoeft, 1967; Hsieh and Bredehoeft, 1987; Roeloffs, 1996; Wang, 2000; Cutillo and Bredehoeft, 2011; Kitagawa et al., 2011; Lai et al., 2013; Wang et al., 2018). This progression in understanding offers a valuable opportunity to evaluate aquifer hydraulic properties through the response of groundwater to tidal fluctuations.

Hsieh and Bredehoeft (1987) introduced the horizontal flow model, focusing on confined conditions influenced by tidal forces. Conversely, Roeloffs (1996) and Wang (2000) explored interactions within vertical flow under tidal fluctuations. Wang et al. (2018) expanded on these studies by incorporating a flow from an upper aquitard, albeit assuming negligible storage within it. Later, Gao et al. (2020) extended these models to include borehole skin effects. Thomas et al. (2023) developed an ET-GW model incorporating storage and strain response in the aquitard. They applied their model to a specific site to evaluate transmissivity variations and validated it using pumping tests.

Numerous studies have investigated aquifer hydromechanical properties by analysing GW level variations induced by Earth tides, employing the models mentioned in the previous literature (Narasimhan et al., 1984; Merritt, 2004; Fuentes-Arreazola et al., 2018; Zhang et al., 2019a; Shen et al., 2020). However, only a limited number of validations have been conducted, which involve comparing the results with robust hydraulic assessments, such as hydraulic conductivity derived from slug testing (Zhang et al., 2019b) or specific storage and transmissivity characterizations obtained through long-term pumping tests (Allègre et al., 2016; Valois et al., 2022). The current evaluations predominantly focus on purely confined conditions, leaving a gap in knowledge regarding tidally induced GW responses in aquifers under semi-confined conditions.

As far as the authors are aware, the publications by Sun et al. (2020), Valois et al. (2022), and Thomas et al. (2023) are the sole references addressing a comparison for a leaky aquifer. Sun et al. (2020) found significant discrepancies between transmissivities obtained from Earth tide fluctuations and those derived from slug or pump tests. However, it is worth noting that the comparison may be subject to discussion, as the authors employed a leaky aquifer model for analysing tidally induced fluctuations, whereas they used a confined aquifer model for conducting slug and pump tests. In the study conducted by Valois et al. (2022), the existing Earth-Tide GroundWater (ET-GW) models, as described





earlier, were unable to reproduce a low semi-diurnal to diurnal amplitude ratio with positive phase
shifts in conjunction with pumping test transmissivity data. This discrepancy highlights the complexity
and challenges in modelling tidally induced groundwater responses in leaky aquifers and the need for
further investigation in this area.
We note that our previous attempts to model the observed substantial amplitude decrease from $O_1$ to
$M_2$ frequency, combined with phase shifts close to zero, proved unsuccessful when using Earth tide
models found in the literature. None of the existing models could provide satisfactory results. The
pursuit of an explanation led to the realisation that analytical solutions with more realistic assumptions
are required. For example, aquifers are widely recognized to be influenced by aquitards, which often
consist of highly porous and compressible clay materials, contributing significant amounts of stored
water to the aquifer (Moench, 1985). Moreover, these aquitards are also impacted by Earth tide strains
(Bastias et al., 2022).
Our first objective is to develop an analytical solution considering storage and strain in the aquitard.
Unlike Thomas et al. (2023), our model incorporates borehole skin effects and allows for a fixed
hydraulic head at the top of the aquitard, broadening its applicability to a broader range of
hydrogeological conditions. The second motivation of our study is to develop a model that better fits
the observed results by considering aquitard storage, as evident in the pumping tests. Third, we
compare the results obtained from our new ET-GW model with those derived from a pumping test in
leaky aquifers with storage in the aquitard. Fourth, since most publications have predominantly
focused on assessing hydraulic properties using the semi-diurnal tide ($M_2$), our third motivation is to
demonstrate the potential of using diurnal tides ($O_1$) in combination with the semi-diurnal ($M_2$ or $N_2$)
to provide a more comprehensive characterization of aquifer and aquitard hydro-mechanical
properties. Our new development offers the potential to enhance hydraulic and geomechanical
subsurface characterization by employing a more realistic model for the groundwater response to
natural forces.


## 2. Groundwater response to Earth tides in a leaky aquifer with aquitard

## storage and strain

Hantush (1960) pioneered the modelling of aquitard storage by modifying the leaky aquifer theory to
account for storage in the aquitard. In our study, we consider a semi-confined configuration (Figure 1)
where the target aquifer is overlain by an aquitard that allows for storage, strain, and vertical flux. Both
layers are assumed to be slightly compressible, spatially homogenous, infinite laterally, and have
constant thickness. Building upon the work of Wang et al. (2018), our research incorporates Earth Tide



(ET) fluctuations into the leaky aquifer equations proposed by Moench (1985). Additionally, we
incorporate the skin effect, as described by Gao et al. (2020).
Groundwater flow and storage in an aquifer overlain by an aquitard can be described as:
$$T\left(\frac{\partial^2 h}{\partial r^2} + \frac{1}{r}\frac{\partial h}{\partial r}\right) = S\left(\frac{\partial h}{\partial t} - \frac{BK_u}{\rho g}\frac{\partial \varepsilon}{\partial t}\right) - K'\frac{\partial h\prime}{\partial z} \tag{1}$$

$$\left(\frac{\partial^2 h\prime}{\partial z^2}\right) = \frac{1}{D\prime}\left(\frac{\partial h\prime}{\partial t} - \frac{B\prime K'_u}{\rho g}\frac{\partial \varepsilon}{\partial t}\right) \quad . \tag{2}$$

Here, $h$ (m) and $h'$ (m) are the hydraulic heads in the aquifer and the aquitard respectively; $h'_j$ (m) is
the fixed hydraulic head at the top of the aquitard, $r$ (m) is the radial distance from the studied well; $T$
(m²/s) and $S$ are the aquifer transmissivity and storativity; $B$, $B'$, $K_u$ (Pa), $K'_u$ (Pa) are the Skempton's
coefficient and the undrained bulk modulus of the aquifer and aquitard respectively; $\rho$ (kg/m³) and $g$
(m/s²) are the water density and gravity constant; $\varepsilon$ is the volumetric Earth tide strain; $K'$ (m/s) is the
aquitard hydraulic conductivity; $S_s'$ (m⁻¹) is the specific storage of the aquitard; $D'$ (m²/s) is aquitard
hydraulic diffusivity ($K'/S_s'$ ratio). Any natural regional groundwater flow is considered negligible.
Borehole drilling causes a zone of damage with a radius $r_s$ (see Figure 1) that is responsible for the skin
effect (Van Everdingen, 1953). A negative skin can be caused by a greater hydraulic conductivity around
the well because of the material damaged by the drilling, while a positive skin can be associated by
porosity clogging caused by the drilling mud. This is reflected in the well's pressure head $\Delta h_s$. The skin
factor ($sk$) can be defined as:
$$sk = \frac{\Delta h_s}{\left(r\frac{\partial h}{\partial r}\right)_{r=r_w}} \quad . \tag{3}$$

Following above assumptions, the boundary conditions are
$$h(r,t) = h_\infty(t) \; at \; r = \infty \tag{4}$$

$$h_w(t) = h(r,t) - sk\left(r\frac{\partial h(r,t)}{\partial r}\right) \; at \; r = r_w \tag{5}$$

$$2\pi r_w T\left(\frac{\partial h}{\partial r}\right)_{r=r_w} = \pi r_c^2 \frac{\partial h_w}{\partial t} \tag{6}$$

$$h' = h \; at \; z = z_i \quad . \tag{7}$$

$$h' = h'_j \; at \; z = 0 \tag{8}$$

Here, t is the time (s); $r_w$ and $r_c$ are the radius of the well screened portion and the radius of well casing
in which water level fluctuates; $z_i$ is the aquifer-aquitard interface elevation (see Figure 1) and $h_w$ is the
hydraulic head at $r_w$.



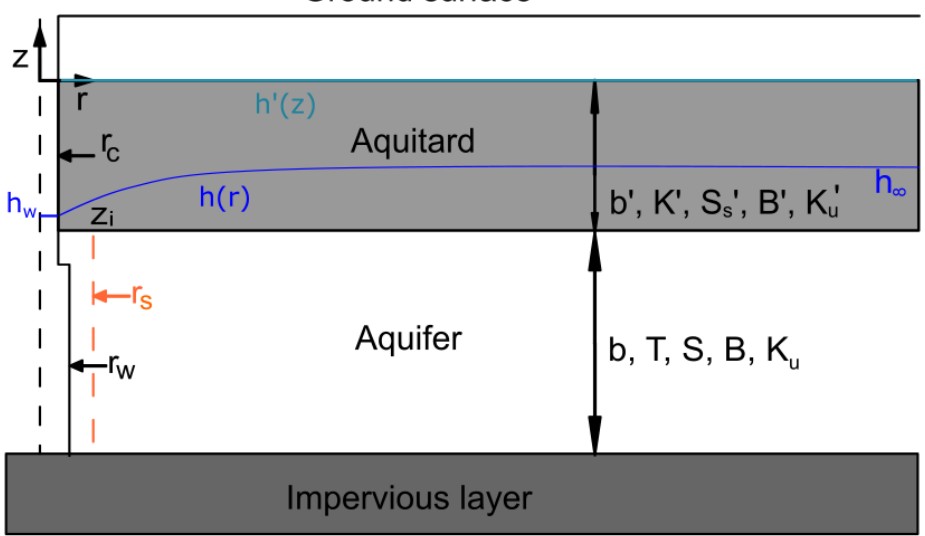


*Figure 1: Semi-confined system with a compressible aquitard with storage*
Following Hsieh et al. (1987) and Wang et al. (2018), complex numbers were used to facilitate harmonic
model development and the solution is obtained by first solving Equation 2 in the aquitard, then
deriving the head response in the aquifer far away from the well ($h_\infty$) which is independent of the
radial distance. Then, the well effect on the aquifer response is considered by using a flux condition at
the well that accounts for wellbore storage. Since $h'$, $h$, $\varepsilon$, $h_w$, $h_\infty$ are all periodic functions, they can be
expressed as:

$$\varepsilon(t) = \varepsilon_0 e^{i\omega t} \qquad (9)$$

$$h_\infty(t) = h_{\infty,0} e^{i\omega t} \qquad (10)$$

$$h_w(t) = h_{w,0} e^{i\omega t} \qquad (11)$$

$$h(r,t) = h_0(r) e^{i\omega t} \qquad (12)$$

$$h'(z,t) = h_0'(z) e^{i\omega t} \quad . \qquad (13)$$

Here, $i = \sqrt{-1}$; $\varepsilon_0$ (m) is the ET strain amplitude and $\omega$ (s$^{-1}$) is the angular frequency. In this case,
Equation 2 becomes:
$$\left( \frac{\partial^2 h_0'}{\partial z^2} \right) = \frac{1}{D'} \left( i\omega h_0' - i\omega \frac{B'K_u'}{\rho g} \varepsilon_0 \right) \qquad . \qquad (14)$$



According to Wang (2000) and Roeloffs (1996) and as detailed in appendix A, the solution of Equation
2 is:
$$h'_0 = A_1 e^{\frac{(1+i)}{\delta}(z-z_i)} + A_2 e^{-\frac{(1+i)}{\delta}(z-z_i)} + \frac{B'K'_u}{\rho g}\varepsilon_0 \qquad (15)$$

where $\delta = \left(\frac{2D'}{\omega}\right)^{1/2}$. Thus, at the interface between the aquifer and the aquitard ($z=z_i$), we have
pressure continuity as $h'(z_i, t) = h_0 e^{i\omega t} = h(t)$ which leads to:
$$\left(\frac{\partial h'}{\partial z}\right)_{z=z_i} = \frac{1+i}{\delta}\left(\frac{h_0 - \frac{B'K'_u}{\rho g}\varepsilon_0}{\tanh\left(\frac{1+i}{\delta}z_i\right)} - \frac{h'_j - \frac{B'K'_u}{\rho g}\varepsilon_0}{\sinh\left(\frac{1+i}{\delta}z_i\right)}\right)e^{i\omega t} \quad . \qquad (16)$$

Equation 1 can be solved far away from the well using $h_\infty$ which is independent of the radial distance
from the well and by using the source term from $h'$ as follows
$$0 = \frac{\partial h_\infty}{\partial t} - \frac{BK_u}{\rho g}\frac{\partial \varepsilon}{\partial t} - \frac{K'}{S}\frac{1+i}{\delta}\left(\frac{h_\infty - \frac{B'K'_u}{\rho g}\varepsilon_0}{\tanh\left(\frac{1+i}{\delta}z_i\right)} - \frac{h'_j - \frac{B'K'_u}{\rho g}\varepsilon_0}{\sinh\left(\frac{1+i}{\delta}z_i\right)}\right)e^{i\omega t} \qquad (17)$$

$$h_{\infty,0} = \frac{BK_u}{\rho g}\varepsilon_0 \frac{Si\omega + K'\frac{1+i}{\delta}\frac{B'K'_u}{BK_u}\left(\frac{-1}{\tanh\left(\frac{1+i}{\delta}z_i\right)} - \frac{h'_j\frac{\rho g}{B'K'_u\varepsilon_0} - 1}{\sinh\left(\frac{1+i}{\delta}z_i\right)}\right)}{Si\omega - K'\frac{1+i}{\delta}\frac{1}{\tanh\left(\frac{1+i}{\delta}z_i\right)}} \quad . \qquad (18)$$

The disturbance in water level due to the well can be expressed as:
$$s(r,t) = h(r,t) - h_\infty(t) \qquad . \qquad (19)$$

Equation 1 becomes:
$$T\left(\frac{\partial^2 s}{\partial r^2} + \frac{1}{r}\frac{\partial s}{\partial r}\right) = S\left(\frac{\partial s}{\partial t}\right) - K'\frac{1+i}{\delta}s\frac{1}{\tanh\left(\frac{1+i}{\delta}z_i\right)} \qquad (20)$$

with the boundary conditions:
$$s(r \rightarrow \infty) = 0 \qquad (21)$$

$$h_{w,0} - h_{\infty,0} = s - sk\left(r\frac{\partial s}{\partial r}\right) \qquad at \; r = r_w \qquad (22)$$

$$2\pi r_w T\left(\frac{\partial s}{\partial r}\right)_{r=r_w} = i\omega\pi r_c^2 h_{w,0} \quad . \qquad (23)$$

The solution of this differential equation is $s(r) = C_I I_0(\beta r) + C_K K_0(\beta r)$ (Wang et al., 2018), where $I_0$
and $K_0$ are the modified Bessel functions of the first and second kind and the zeroth order, respectively.
Further,



$$\beta = \left( \frac{i\omega S}{T} - \frac{K'}{T} \frac{(1+i)}{\delta} \frac{1}{\tanh\left(\frac{1+i}{\delta} z_i\right)} \right)^{1/2} \quad . \qquad (24)$$

The boundary conditions lead to $C_I=0$ and $C_K = -\frac{i\omega r_c^2 h_{w,0}}{2T\beta r_w K_1(\beta r_w)}$ because $\frac{dK_0(r)}{dr} = -K_1(r)$. Therefore,
the final solution for the well water level is:
$$h_{w,0} = \frac{BK_u}{\rho g} \varepsilon_0 \frac{Si\omega + K'\frac{1+i}{\delta}\frac{B'K_u'}{BK_u}\left(\frac{-1}{\tanh\left(\frac{1+i}{\delta}z_i\right)} - \frac{h'\frac{\rho g}{JB'K_u'\varepsilon_0}-1}{\sinh\left(\frac{1+i}{\delta}z_i\right)}\right)}{\sigma\left(Si\omega - K'\frac{1+i}{\delta}\frac{1}{\tanh\left(\frac{1+i}{\delta}z_i\right)}\right)} \quad , \qquad (25)$$

where
$$\sigma = 1 + \frac{i\omega r_c^2 K_0(\beta r_w)}{2T\beta r_w K_1(\beta r_w)} + \frac{i\omega r_c^2}{2T} sk \, . \qquad (26)$$

By assuming $h_j' = 0$ (i.e. the hydraulic head at the top of the aquitard corresponds to the
unsaturated-saturated interface at z=0), Equation 25 can be reorganized to
$$h_{w,0} = \frac{BK_u}{\rho g} \varepsilon_0 \frac{Si\omega + K'\frac{1+i}{\delta} R_{KuB}\left(\frac{1-\cosh\left(\frac{1+i}{\delta}z_i\right)}{\sinh\left(\frac{1+i}{\delta}z_i\right)}\right)}{\sigma\left(Si\omega - K'\frac{1+i}{\delta}\frac{1}{\tanh\left(\frac{1+i}{\delta}z_i\right)}\right)} \quad , \qquad (27)$$

where
$$R_{KuB} = \frac{K_u'B'}{K_uB} \quad . \qquad (28)$$

By disregarding $\frac{BK_u}{\rho g}$ product which only controls the amplitude, the solution has six independent
parameters which are $T$, $S$, $K'$, $D'$, $sk$ and the $R_{KuB}$ ratio.
Let us now define the amplitude response (or amplitude ratio), $A$, and phase shift, $\alpha$, of the GW
response to ET fluctuations:
$$A = \left| h_{w,0} \middle/ \frac{BK_u}{\rho g} \varepsilon_0 \right| \qquad (29)$$

$$\alpha = arg \left[ h_{w,0} \middle/ \frac{BK_u}{\rho g} \varepsilon_0 \right] . \qquad (30)$$

Figure 2 shows the amplitude response and phase shift as a function of frequency using our new
solution in comparison to key models reported in the literature. Aquitard parameters were set accoring
to Batlle-Aguilar et al. (2016), while aquifer parameters were chosen accoring to the field application
below. We validate the solution using a very low aquitard conductivity ($10^{-14}$ m/s), so that we can



compare it to the horizontal flux with wellbore storage model (Hsieh et al., 1987). It shows a perfect
match.
Because both horizontal and vertical flux models are associated with opposite phase shift signs (Figure
2b), the latter can offer valuable insights for model selection (Allègre et al., 2014). Positive phase shifts
in the vertical flux model are related to an increasing amplitude ratio with frequency, whereas the
wellbore storage model exhibits the opposite behaviour. Wang et al. (2018) developed a leaky model
capable of demonstrating both positive and negative phase shifts, where positive phase shifts
correspond to an increasing amplitude ratio with frequencies, and negative phase shifts are linked to
a decreasing amplitude ratio.
Our new model showcases positive or negative phase shifts with an increasing or decreasing amplitude
ratio over frequency, even allowing for amplitude ratios greater than one. Notably, Wang (2000)
observed a similar characteristic in the vertical flux model, with an amplitude just above one (1.06) for
very specific conditions. At high frequencies, our model displays amplitude ratios and phase shifts
similar to those of the leaky and wellbore storage models, reflecting the attenuation of high-frequency
pore pressure fluctuations in the aquifer by well water.



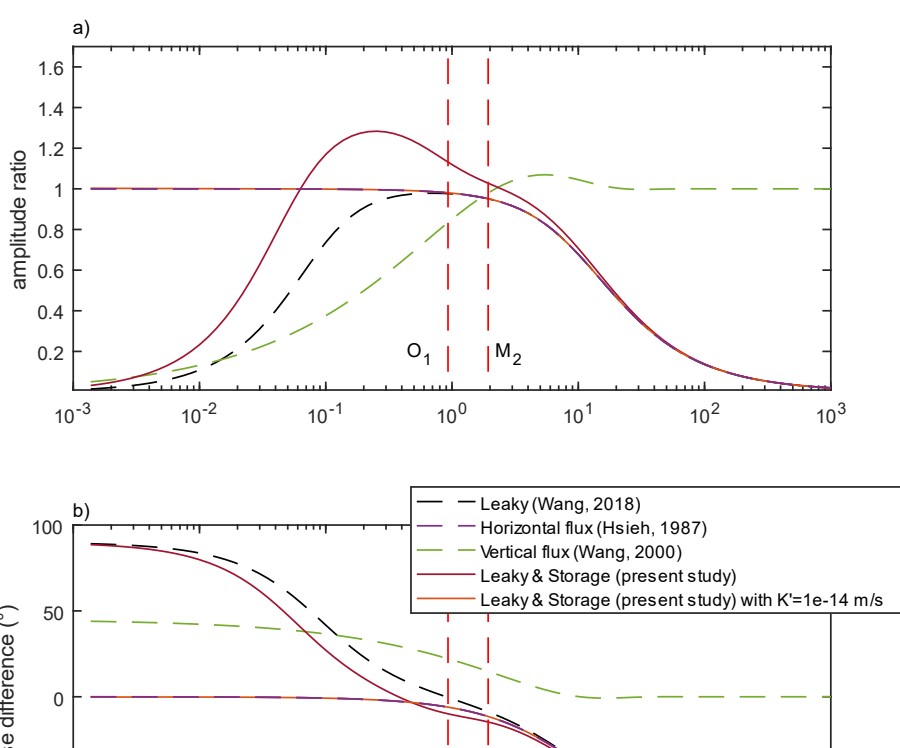


*Figure 2: Frequency variation of amplitude response and phase shift of the groundwater response to Earth tides. The transmissivity (T) is $10^{-5}$ m²/s, storativity (S) is $10^{-4}$, hydraulic conductivity of the aquitard (K') is $10^{-8}$ m/s, aquitard hydraulic diffusivity (D') is $10^{-4}$ m²/s, skin factor (sk) is 0, $R_{KuB}$ to 1.4, well casing radius ($r_c$) and screen radius ($r_w$) is 6.03 cm. b' was set to 5 m. Screen depth (z) was set to 23 m for the vertical flow model of Wang (2000).*


We explored the parameter space by focussing on the frequency-dependant amplitude response and
phase shift responses for different sets of parameter values. The reference parameter set is the one
described above. $T$, $R_{KuB}$ and $z_i$ have a large impact on model shapes. As also observed by Hsieh et al.
(1987), $S$ does not have a mojor impact on the results (Fig3c and 3d). The skin effect does not play a
large role in the useful frequency band for amplitudes, but its influence is larger for the phase shifts
when compared to the reference parameter set. $K'$ does not significantly influence the results with



respect to the reference parameter set used in the study (Figures 3e to 3f). $K'$ has the opposite role of
$S$ and they appear to compensate each others effects, because of their respective role in Equation 27.

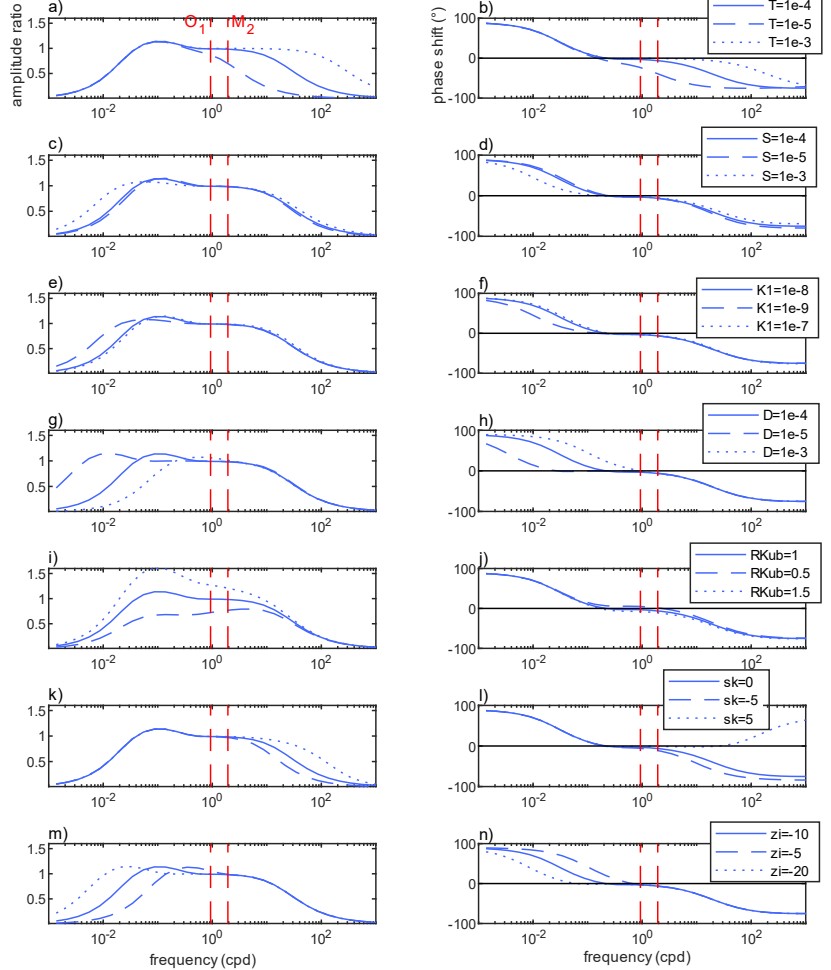


*Figure 3: Illustration of the amplitude response (left column) and phase shift (right column) as a function of frequency for*
*various parameters compared to the reference parameter set.*





**3. Application of the new model to a groundwater monitoring dataset from Cambodia**

  **3.1 Field site and previous results**

The field site in Northwest Cambodia comprises three boreholes drilled into the subsurface, consisting of mudstone, claystone, siltstone, and sandstone. Time series and pumping test results have been reported in Valois et al. (2022), while details of the lithology can be found in Vouillamoz et al. (2012; 2016) and Valois et al. (2017; 2018; 2022). Pumping from the aquifer is limited by a very low specific yield, attributed to the presence of fine deposits such as clay and mudstone (Vouillamoz et al. 2012; Valois et al., 2018).

The boreholes were drilled to a depth of 31 meters with a radius of 6 inches and equipped with 4-inch PVC casing from top to bottom, featuring a 9-meter long screen at the hole's base. The aquifer is situated within a hard rock media, comprising either claystone or sandstone, located beneath a 10-meter thick clay layer.

For the pumping tests, the wells were pumped for three days, and water levels were allowed to recover for four days in two observation wells. The interpretation of the pumping tests utilized AQTESOLV™/Pro v4.5 software, employing the leaky aquifer with aquitard storage model (Moench, 1985) or a 3D flow using the generalized radial flow model (Barker, 1988). The selected solutions, compared to other models (Theiss, Hantush without aquitard storage), demonstrated the best fit with a Root Mean Square Error (RMSE) of 0.02 m for Cambodia.

**3.2 Well sensitivities and phase shifts to Earth tides**

Between 2010 and 2015, well water levels were measured at 20, 40, or 60-minute intervals using absolute pressure sensors (Diver data loggers, Eijkelkamp Soil & Water, NL). To compensate for barometric pressure (BP) effects, data from a barometer located a few kilometres away from the field site were utilized (Eijkelkamp Soil & Water, NL). A zero-phase Butterworth filter was employed to eliminate low-frequency content (periods longer than 10 days) from both groundwater (GW) and BP data.

For each site's geolocation (latitude, longitude, and height), ET strain time series were computed at 20-minute intervals using SPOTL software (Agnew, 2012). The time series were then modelled using Harmonic Least-Squares (HALS; Schweizer et al., 2021) with eight frequencies corresponding to the major tides (Table 1) following Merritt's description (2004). HALS provides amplitude and phase estimations for each tidal component and record.





**Table 1.** *Dominant tidal components that are generally found in groundwater measurements*
*(adapted from MacMillan et al., 2019)*

| Darwin Name | Frequency (cpd) | Attribution |
|---|---|---|
| $Q_1$ | 0.89365 | Earth |
| $O_1$ | 0.929536 | Earth |
| $P_1$- | 0.997262 | Earth |
| $S_1$ | 1.000000 | Atmosphere |
| $K_1$ | 1.002738 | Earth |
| $N_2$ | 1.895982 | Earth |
| $M_2$ | 1.932274 | Earth |
| $S_2$ | 2.000000 | Earth/Atmosphere |
| $K_2$ | 2.005476 | Earth |


The results obtained from HALS were utilized to calculate the amplitude response and phase shift
between groundwater (GW) and Earth tide (ET) for each tidal component. These amplitude responses
are commonly known as "well sensitivities" to Earth tide strains (Rojstaczer and Agnew, 1989) and are
summarized in Figure 4 alongside the corresponding phase shifts.
The well sensitivities to tides exhibit a frequency-dependent behaviour, resulting in similar values for
neighbouring frequencies and a generally decreasing magnitude. The amplitudes of $M_2$ and $N_2$ are
relatively straightforward to assess due to their significant magnitudes (11.2 and 2.1 mm, respectively),
and their amplitude responses and phase shifts are highly similar. The phase shifts for the tides of
interest ($O_1$, $N_2$, and $M_2$) are positive. However, the signs of the amplitudes for the other tides can be
attributed to their low amplitude responses, which are challenging to characterize using HALS.



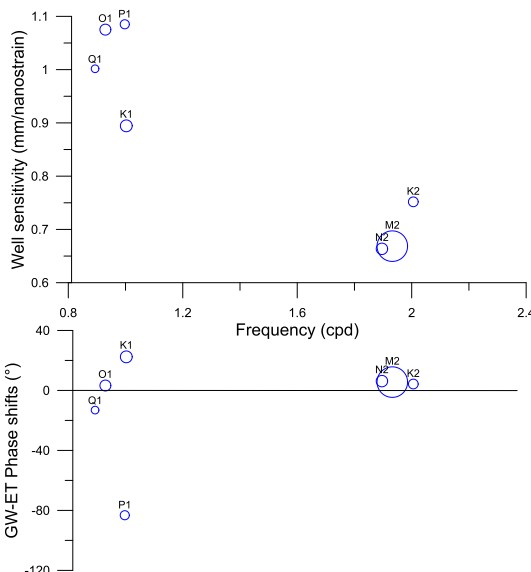

*Figure 4: Amplitude responses and phase shifts as a function of frequency for the Cambodian site. S1 and S2 were excluded as they are not only generated by Earth tides. The circle size is proportional to the amplitude in the well water levels.*

### 3.3 Fitting the $M_2$/$O_1$ amplitude response ratio and phase shifts

The analysis is restricted to two types of tides: the semi-diurnal and diurnal tides. This limitation arises because the magnitude of Earth tide-induced well water levels is significantly damped for higher frequencies, making it difficult to discern and analyse tides beyond these two types. Here, we use amplitude responses ($A_{M2}$, $A_{O1}$) and phase shifts ($\alpha_{M2}$, $\alpha_{O1}$) to estimate hydraulic subsurface properties. $N_2$ tide was not used since its response may be too similar to $M_2$ and does not help with constraining the model. Amplitudes are influenced by geomechanical parameters ($BK_u$) which are generally not considered in classical hydrogeology. Valois et al. (2022) previously illustrated that the $M_2$ to $O_1$ amplitude response ratio can be computed because it is not directly multiplied by $BK_u$ and because it provides useful information about model choice. This leads to a system of three equations and six parameters ($T$, $S$, $K'$, $D'$, $skin$, and $R_{KuB}$) by using the simplified model in Equation 27 when the geometry of the well and the aquitard-aquifer system is known:

$$\frac{A_{M2}}{A_{O1}} = \left| h_{w,0,\omega=M_2} / h_{w,0,\omega=O_2} \right| \qquad (31)$$

$$\alpha_{M2} = arg\left[ h_{w,0,\omega=M_2} / \frac{BK_u}{\rho g} \varepsilon_0 \right] \qquad (32)$$

$$\alpha_{O1} = arg\left[ h_{w,0,\omega=O_1} / \frac{BK_u}{\rho g} \varepsilon_0 \right]. \qquad (33)$$

Table 2 displays the data to be fitted using the three equations above.



*Table 2: Data to be fitted using the ET-GW model*

|  | $\dfrac{A_{M2}}{A_{O1}}$ | $\alpha_{M2}$ (°) | $\alpha_{O1}$ (°) |
|---|---|---|---|
| Cambodia | 0.62 | 5.62 | 3.3 |


A systematic exploration of the entire parameter space without any constraints other than the well
and aquifer geometry was carried out. Hydraulic and geomechanical property ranges are chosen
according to the literature, i.e., De Marsily (1986) and Domenico and Schwartz (1998). In order to
assess the goodness of fit with the three observed parameters (Eqs 31 to 33), the objective chi-square
function is defined below:
$$\chi^2 = {}^1/_N \sum_{i=1}^{N} \left( \frac{(Obs_i - Mod_i)}{Error_i} \right)^2 \quad (34)$$
where $N$ is the number of observed parameters (3 here), $Obs_i$, $Mod_i$ and $Error_i$ are the observed
parameter, modeled parameter and their errors respectively. Thus, this objective function takes into
account errors of the observed parameters (De Pasquale et al.; 2022). They were set to 0.1°, 0.5° and
0.2 for $\alpha_{M2}$, $\alpha_{O1}$ and $\frac{A_{M2}}{A_{O1}}$ respectively, according to Valois et al. (2022).

The model allows to fit both $O_1$ and $M_2$ positive phases and the low $M_2$ to $O_1$ amplitude ratio (misfit
closed to 0 in Fig. 5), whereas the model of Gao et al (2020) cannot (misfit above 1 see Appendix B,
and Valois et al., 2020). The $T$ value is in good agreement with the pumping test range (Fig. 5a). $S$ is
half an order of magnitude below the pumping test range (Fig 5b) whereas the storativity best-fit for
Gao et al (2020) is two orders of magnitude above (Appendix B). The skin effect also shows acceptable
values as compared to the pumping test (Fig 5f). The parameter exploration shows best-fits for $K'$ and
$D'$, whereas it is difficult to identify a clear best-fit for the $R_{KuB}$ parameter. The values are within the
expected range for the hydrogeological configuration: The mudstone aquitard has a lower hydraulic
conductivity ($10^{-8}$ m/s) than the underlying claystone aquifer (coarser grain size than the aquitard, with
a $K$ value of about $10^{-7}$ m/s for an aquifer thickness of 22 m), and a diffusivity of about $10^{-4}$ m²/s. This
is in agreement with the aquitard classification of Pacheco (2013).



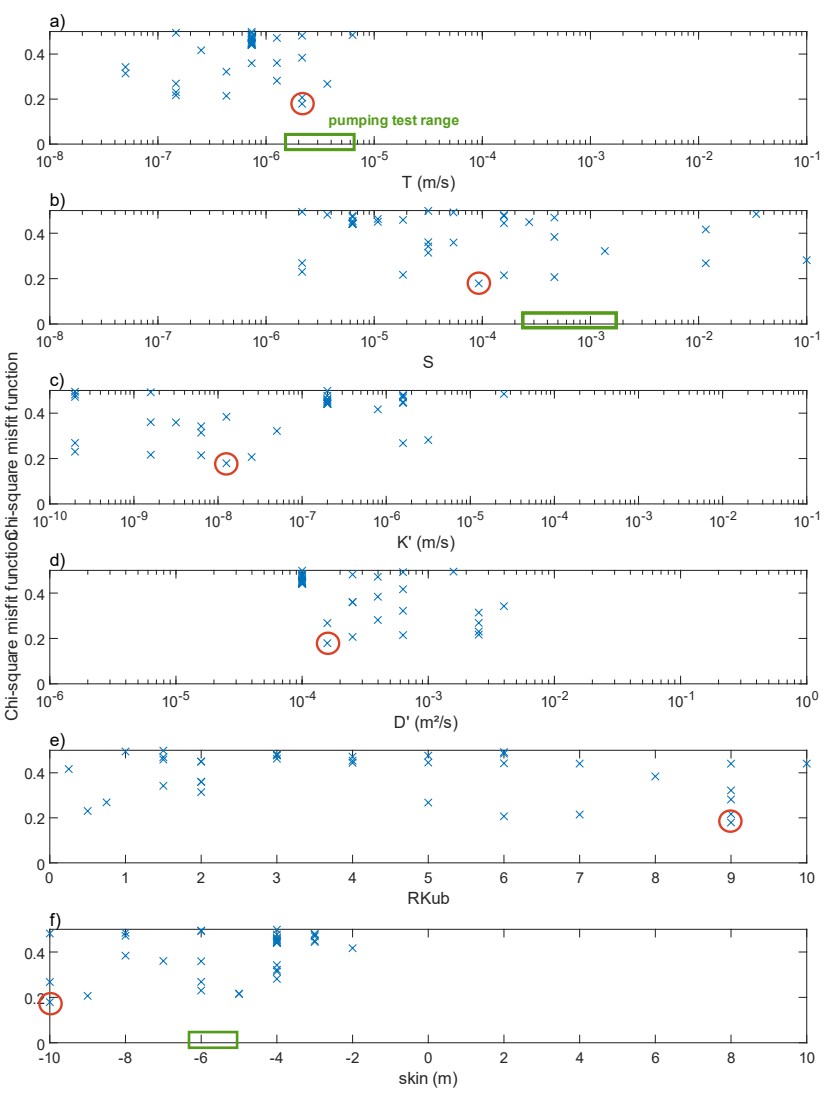


Figure 5: Results of full parameters exploration using two leaky aquifer models for the Cambodian case study.


## 4. DISCUSSION

### 4.1. Uncertainties and discrepancies

There are several sources of uncertainty which originate from measurement and their propagation as well as uncertainties introduced by model assumptions. We believe that uncertainties linked to pressure sensor resolution (0.2 mm) and time resolution (20 minutes) as well as the HALS

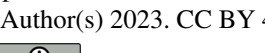



decomposition were too low to be worth considering, at least for the semi-diurnal tides. This can be
deduced from the nearly identical amplitude responses at $M_2$ and $N_2$ at our field site. Because those
responses are indeed identical, it means that errors in the raw data set did not influence the response
characterization. We note that amplitude responses and phase shifts show larger discrepancies for the
diurnal tides. This could be linked to overall lower amplitudes which are generally more difficult to
characterise. We therefore conclude that errors arising from uncertainties are negligible compared to
the uncertainty introduced by model assumptions, in agreement with Sun *et al.* (2020).
Discrepancies between hydro-geomechanical properties derived from the groundwater response to
Earth tides (termed as "passive" and assuming a compressible matrix) and hydraulic testing (e.g., slug,
pump and lab testing, termed as "active" and generally assuming an incompressible matrix) have been
reported in the literature and have not been appropriately reconciled. By fitting amplitude response
ratio and phase shifts (Section 3.3), a T value discrepancy of one order of magnitude can be observed
between both approaches. We hypothesise that this is caused by parameter anisotropy.
Zhang *et al.* (2019b) pointed out differences in hydraulic conductivities of more than one order of
magnitude between ET analysis and slug tests and attributed this to differences in the investigated
scale. Allègre *et al.* (2016) reported much higher values of storativity derived from pumping test is
compared to ET when using the vertical flow model. Sun *et al.* (2020) showed that *T* values are
frequency-dependent with several orders of magnitude differences when comparing co-seismic, ET,
slug or pump test methods. The discrepancies can be explained by the different conceptual models
used in the active (based on perfectly confined) and passive methods (based on leaky conditions) or
by the frequency dependency of hydraulic parameters. The literature illustrates that transmissivity,
hydraulic conductivity or specific storage can indeed vary depending on the frequency of the forcing
(e.g., Cartwright *et al.,* 2005; Renner and Messar, 2006; Guiltinan and Becker, 2015; Rabinovich *et al.,*
2015). This demonstrates the need for attention when assessing hydraulic parameters using passive
methods for semi-confined conditions. We specifically emphasise the need for using the same
conceptual model (i.e., confined, leaky with or without storage, vertical flow) when comparing active
and passive methods, as well as the need of preliminary hydrogeological knowledge of both the aquifer
system (i.e., presence of an aquitard with or without storage) (Bastias et al., 2022) as well as the
borehole skin effect.

**4.2. The use of the leaky model with aquitard storage**
Our new analytical solution describing the well water level response to harmonic Earth tide strains
contains at least six hydro-geomechanical parameters that could be derived from only three features,
e.g., $M_2$ to $O_1$ amplitude response ratio and $M_2$ and $O_1$ phase shifts. Applying this model to real-world





cases to derive properties from amplitude responses and phase shifts provides relevant information
on $T$, $S$, $D'$, $K'$, and skin effect, but it is prone to non-uniqueness. Thus, a priori information may be
needed depending on the capacity of the inverse problem to fit observed data (phases shifts and
amplitude ratio). In our case study, parameter assessment would benefit from prior information on $S$
(or $K'$) and $R_{KuB}$.
The model presented in this study can be useful when the hydrogeological configuration involves
storage in the aquitard with fixed head (i.e., Dirichlet) boundary conditions and for cases where phase
shifts and amplitude ratio do exemplify a specific pattern. For example, when compared to Gao et al
(2020) and using the parametrisation of the present study (Fig. 6), our solution is able to model lower
$M_2$ to $O_1$ amplitude ratio, lower phases, and higher amplitude ratio for phases closed to 0.

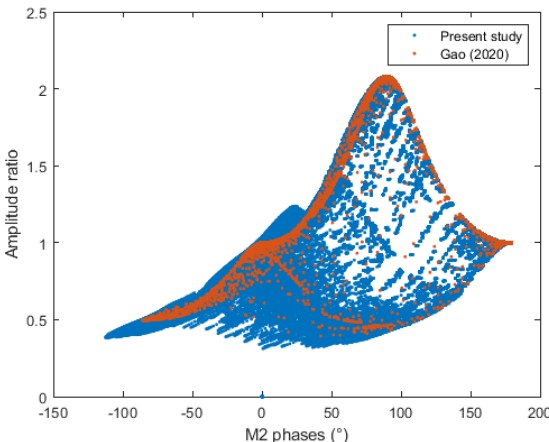


*Figure 6: Outputs of the models using the parametrization of the study (rc=rw=6.08 cm, $z_i$=-10 m).*
While the amplitudes are controlled by the product of the Skempton coefficient and the undrained
bulk modulus, these mechanical parameters also affect phase shifts. Therefore, further investigations
are needed to assess these influences using other methods or to link them empirically with the
hydraulic parameters. This is crucial to enhance confidence in utilizing groundwater response to Earth
tides as a valuable tool for better understanding and characterizing groundwater resources.

## 5. Conclusion

We have developed a new analytical solution for the well water level response to Earth tide strains.
This solution considers a previously unprecedented physical reality, specifically, a leaky aquifer with



aquitard storage, subject to Dirichlet boundary conditions under tidal strain. Additionally, our model
considers the influence of borehole storage and skin effects, further improving the accuracy and
comprehensiveness of the analysis. This model extends upon previous models and allows advanced
characterization of the subsurface using the groundwater response to natural forces. The new model
overcomes previous limitations, for example it explains very low $M_2$ to $O_1$ amplitude ratios as well as
large phase shift difference between $M_2$ and $O_1$ tides. The model relies on six combinations of hydro-
geomechanical parameters. In this study, we assess the most sensitive parameters to be the
transmissivity, the well skin effect, the aquitard to aquifer mechanical parameters ratio ($B'K_u'/BK_u$), as
well as aquitard diffusivity and aquitard conductivity to aquifer storativity ratio.
We apply our new model to a groundwater monitoring dataset from Cambodia and compare the
results with pumping tests undertaken in the same formation. We used the diurnal ($O_1$) and semi-
diurnal ($M_2$) tides to better constrain the model. Results illustrate significant insight into subsurface
properties. For example, we derive relevant information about $T$, $S$, $D'$, $K'$, and *skin effect*, when
compared to the pumping test results. Overall, our new model can be used to shed light on previously
inexplicable well water level behaviour and can be paired with other investigation methods to enhance
understanding of subsurface processes.

## Competing interests

The contact author has declared that none of the authors has any competing interests.

## Acknowledgements

This work has been carried out in the framework of the Institut de Recherche pour le Développement
and the French Red Cross collaborative project 39842A1 - 1R012-RHYD, with the financial support of
the European Community (grant DIPECHO SEA ECHO/DIP/BUD/2010/01017 and grant DCI-
FOOD/2011/278-175).

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



## Appendix

**Appendix A: The analytical solution in the aquitard**

To solve Equation 14 with the boundary conditions in Equations 7 and 8, we define:

$$\widehat{h_0'} = h_0' - \frac{B'K_u'}{\rho g}\varepsilon_0 \qquad\qquad \text{(A1)}$$

Thus, the equation system become:

$$\left(\frac{\partial^2 \widehat{h_0'}}{\partial z^2}\right) = \frac{i\omega \widehat{h_0'}}{D'} \qquad\qquad \text{(A2)}$$

$$\widehat{h_0'}(z = z_i) = h_0 - \frac{B'K_u'}{\rho g}\varepsilon_0 \qquad\qquad \text{(A3)}$$

$$\widehat{h_0'}(z = 0) = h_j' - \frac{B'K_u'}{\rho g}\varepsilon_0 \qquad\qquad \text{(A4)}$$

The solution $\widehat{h_0'}$ is of the form:

$$\widehat{h_0'} = A_1 e^{\frac{(1+i)}{\delta}(z-z_i)} + A_2 e^{-\frac{(1+i)}{\delta}(z-z_i)} \qquad\qquad \text{(A5)}$$

It yields

$$A_1 = \frac{e^{\frac{(1+i)}{\delta}z_i}\left(h_0 - \frac{B'K_u'}{\rho g}\varepsilon_0\right) - \left(h_j' - \frac{B'K_u'}{\rho g}\varepsilon_0\right)}{2\sinh\left(\frac{(1+i)}{\delta}z_i\right)} \qquad\qquad \text{(A6)}$$

$$A_2 = \frac{-e^{-\frac{(1+i)}{\delta}z_i}\left(h_0 - \frac{B'K_u'}{\rho g}\varepsilon_0\right) + \left(h_j' - \frac{B'K_u'}{\rho g}\varepsilon_0\right)}{2\sinh\left(\frac{(1+i)}{\delta}z_i\right)} \qquad\qquad \text{(A7)}$$

Thus:

$$h_0' = A_1 e^{\frac{(1+i)}{\delta}(z-z_i)} + A_2 e^{-\frac{(1+i)}{\delta}(z-z_i)} + \frac{B'K_u'}{\rho g}\varepsilon_0 \qquad\qquad \text{(A8)}$$

**Appendix B: Additional information on parameter exploration**

The figure A1 shows the misfit functions using the model developed in this study and the model of
Gao et al . (2020). Misfits are clearly higher for the older model that do not consider storage and
tidal response in the aquitard. Storativity best-fit using the model of Gao et al (2020) failed to
reproduce pumping test values. Nevertheless, transmissivity and skin estimates fall within
pumping test range.



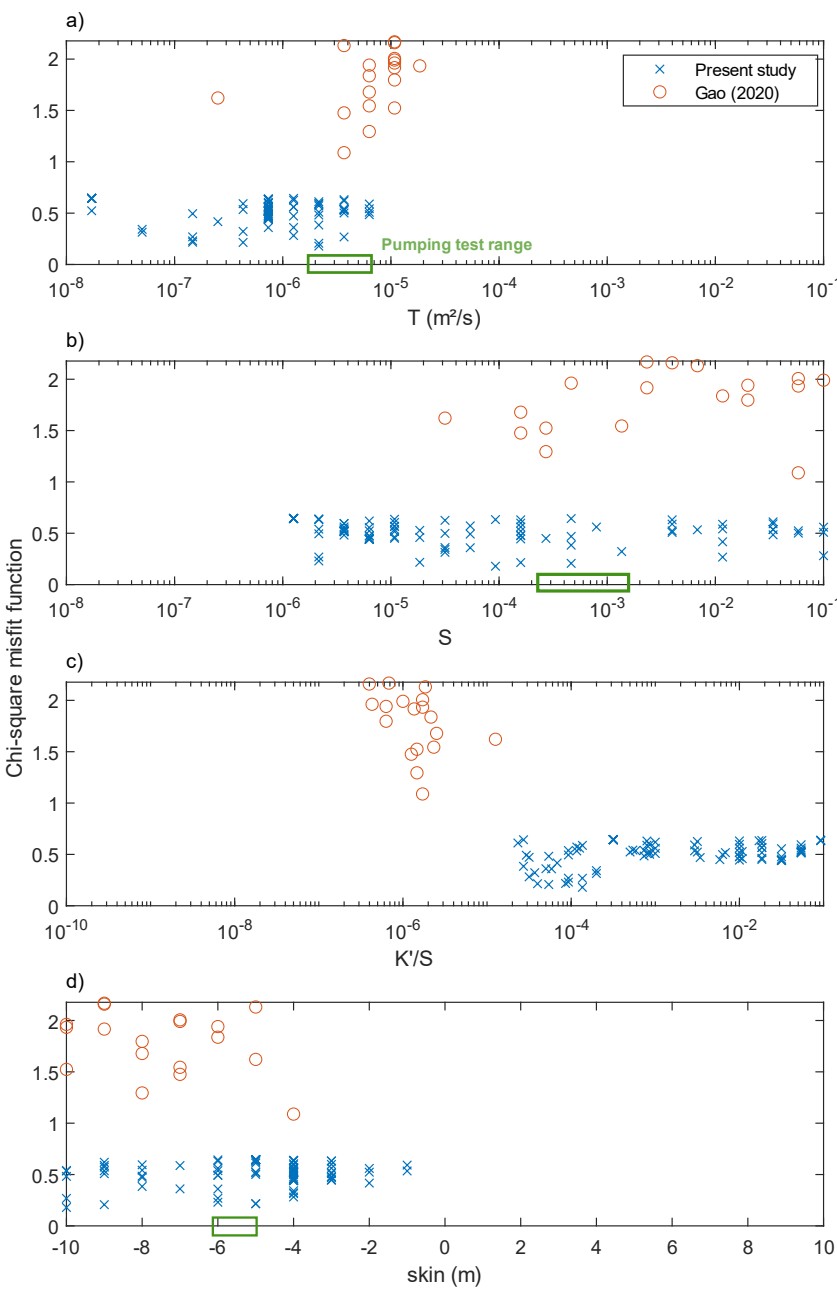


*Figure A1: Comparison of misfit function for the present model and the one of Gao et al. (2020) for the Cambodian case study.*
*Only the first hundred best-fit were plotted.*