# Peer review of "Technical Note: Analytical Solution for Well Water Response to Earth Tides in Leaky Aquifers with Storage and Compressibility in the Aquitard"

_EGUsphere, 2023_

## Author Comment (AC1)

**Reviewer 1**

General comments

This paper presents a new analytical solution for earth tide induced well water level fluctuations in an aquifer and an aquitard. A cylindrical coordinate system is utilized with an infinite radius while the thickness of the aquifer and aquitard are defined manually. The periodic earth tide is expressed by amplitude and angular frequency. The governing equation of the hydraulic head in the aquitard is solved first. Its solution is then used to solve the governing equation for the aquifer.

The key contribution is that the model considers the borehole skin effects in a leaky aquifer. My recommendation is major revision.

Key general comments are:

1. A key concern is the applicability of the 2-D cylindrical coordinate which describes the well level fluctuation with earth tide source far away from the observation well. If this description is correct, the conceptual model can be simplified as a cylinder with a source at the outer surface and an observation well at the central line. In this case, the model is more suitable for an island or peninsula setting. While the inland observation well in Cambodia may not be this case. Please justify the insignificance of the angular coordinate in the solution.

   I think there is a misunderstanding, the source is not at the outer surface with oceanic tides, but everywhere in the geological media as Earth tides. In fact, earth tides provoke a harmonic strain field (compression and dilatation) everywhere in Earth. So that the aquifer and aquitard are subject to this strain. This is the reason why 2-D cylindrical coordinates are used, because of the radial symmetry of the well surrounded by the source. This type of configuration is common in Hydrogeology (De Marsily, 1986) and in aquifer fluctuations from earth tides forces (Hsieh, 1986; Wang, 2000 ; Gao et al, 2020).

   Thus, we will modify the introduction (last sentence of the first paragraph, first sentence of the 2nd paragraph, first sentence of the last paragraph) in the introduction section to clarify the role of Earth tides. We will add also "2D cylindrical coordinates are used because of the radial symmetry caused by the well surrounded by hydrogeological material affected by tidal strain" at the end of the 1st paragraph in section 2. We will also add figure 4 that better explains the processes involved.

   References :

   De Marsily, G. (1986). *Quantitative hydrogeology*. Paris School of Mines, Fontainebleau.

   Gao, X., Sato, K., & Horne, R. N. (2020). General Solution for Tidal Behavior in Confined and Semiconfined Aquifers Considering Skin and Wellbore Storage Effects. Water Resources Research, 56(6), e2020WR027195.

Hsieh, P. A., Bredehoeft, J. D., & Farr, J. M. (1987). Determination of aquifer transmissivity from Earth tide analysis. Water resources research, 23(10), 1824-1832.

Wang, H. F. (2000). Theory of linear poroelasticity with applications to geomechanics and hydrogeology (Vol. 2). Princeton University Press.

2. To better understand the physics of the water level fluctuations in response to the tide, please plot the hydraulic head distribution contour in the results section for several time snapshots to show the effect of skin effect and the leaky aquifer.

Because of the radial symmetry, as explained above, we did not plot the hydraulic head distribution contours. We could plot the hydraulic head distribution with respect to the radial distance, but because the skin effects are calculated for the hydraulic head in the well, the calculation is independent of the radial distance. This is the reason why we choose to plot rather the hydraulic head time series (Fig 4c) as compared to the volumetric tidal strain source (Fig 4a), and to compare conditions with or without skin effects to better show the impacts of skin effects (phase shift in Fig 4c). Regarding the impacts of a compressible leaky aquifer, we choose to follow the same representation by comparing purely confined and leaky conditions for the hydraulic head time series within the aquifer (phase shift too in Fig 4b).

We will add at the end section 2 the following paragraph and the new figure 4

"Figure 4 shows the impacts of considering leaky and skin effects for a realistis parameter set. Purely confined conditions (no leaky aquitard) do not create a phase shift between the volumetric strain in the aquifer (Fig 4a) and the associated hydraulic head variation (Fig 4b), while a compressible and leaky aquitard (Fig 4b) or skin effects around the well (Fig 4c) could involve positive phase shifts. "

[Figure]

*Figure 1: Example of the volumetric strain time series generated by the M2 Earth tide in a), which creates aquifer hydraulic head variations in b), resulting in well water level variations in c). The transmissivity (T) is $10^{-6}$ m²/s, storativity (S) is 7 $10^{-4}$, hydraulic conductivity of the aquitard (K') is $10^{-6}$ m/s, aquitard hydraulic diffusivity (D') is $10^{-4}$ m²/s, skin factor (sk) is -5 m, $z_i$ to -10 m, RKuB to 0.3, well casing radius (rc) and screen radius (rw) is 6.03 cm. B is set to 0.8 and $K_u$ to 10 GPa.*

3. Add more relevant references including the solutions for fractured rock aquifers.

   We will add in the 3rd paragraph in the Introduction some relevant references:

Bower, D. R. (1983). Bedrock fracture parameters from the interpretation of well tides. *Journal of Geophysical Research: Solid Earth*, 88(B6), 5025-5035.

Burbey, T. J., Hisz, D., Murdoch, L. C., & Zhang, M. (2012). Quantifying fractured crystalline-rock properties using well tests, earth tides and barometric effects. *Journal of Hydrology*, 414, 317-328.

Carr, P. A., & Van Der Kamp, G. S. (1969). Determining aquifer characteristics by the tidal method. *Water Resources Research*, 5(5), 1023-1031.

Rahi, K. A., & Halihan, T. (2013). Identifying aquifer type in fractured rock aquifers using harmonic analysis. *Groundwater*, 51(1), 76-82.

Sedghi, M. M., & Zhan, H. (2016). Hydraulic response of an unconfined-fractured two-aquifer system driven by dual tidal or stream fluctuations. *Advances in water resources*, *97*, 266-278.

4. Please share the field data being used in this study if applicable.

We uploaded the data in :

https://github.com/remival/CambodiaData-for-leaky-and-compressible-aquiatrd.git

We will add the link in the data availability section

Line 15-21: The key points list is not in standard form, with phrases and sentences mixed. For example, "Derivation of semi-analytical solutions for equivalent permeability in fractured multilayered porous media." Please check the HESS articles as an example.

We checked some HESS papers, and it seems there is no key points in this journal. Nonetheless, we will modify the existing key points to have only sentences.

Line 65: Consider adding references that apply leaky aquifers in pumping tests, e.g.,

Butler and Tsou 2003 (https://doi.org/10.1029/2002WR001484),

Wen et al. 2011 (https://doi.org/10.1016/j.jhydrol.2011.01.010),

Ok, references will be added in the text (after equation 16 and in the introduction to defines the skin factor)

Line 76-77: Please describe what O1 and M2 represent when they appear the first time. Consider moving the 'semi-diurnal' and 'diurnal tide' in lines 91 and 92 to lines 76 and 77.

We will do as suggest

Line 108: Cite a classic reference for the governing equations (1) and (2).

We will add De Marsily (1986) reference for equations (1) and (2)

Line 123: Also give a reference that defines the skin factor.

Wen et al. 2011 will be cited to define the skin factor.

Line 137, 155, 159: "then deriving the head response in the aquifer far away from the well ($h\infty$)". I am confused by the expression. Is there another well at $h\infty$ or just a hydraulic head at a specific position at r= ?

There is no other well. $h\infty$ represents the hydraulic head in the aquifer (not inside a well that could impact amplitude and phase harmonics from the aquifer) far from the well. Thus, we will add (above equation 9):

"$h_\infty$ is the aquifer response to the tidal harmonic sources far from the well. Thus, $h_\infty$ is the aquifer hydraulic head response without any disturbance from a well-aquifer system."

Line 162: How is equation 20 derived? Could you give more details or explanations?

To obtain such an equation, we combined Equation 19 with Equation 1 to express equation 1 with $s+h_\infty$. We used Equation 16 to express the leaky flow $\left(\frac{\partial h\prime}{\partial z}\right)$, and we used equation 17 to remove the far field component ($h_\infty$). Thus we obtained Eq 20.

We will add above Eq 20 :

"By expressing Equation 1 with the sum of $s$ and $h_\infty$ (Eq. 19), and using Equation 16 to express the leaky component and using equation 17 to remove $h_\infty$, it follows:"

Line 207: Please increase the line width of the dashed lines in Figure 2. It is difficult to tell the difference between the solid lines and dashed lines when they are well-fitted and printed in black-and-white mode.

OK, we will increase the line width, so that it easier to see the differences between the solid lines and dashed lines.

[Figure]

Line 216: Could you specify 'useful frequency band'?

The useful frequency band is about 1 to 2 cpd, where Earth tides can be noticed in the groundwater level time series.

We will add details about the useful feequency band in the text (section 2).

Line 227: Consider adding references related to groundwater flow in fractured rock.

We will add references related to groundwater flow in fractured rock :

Bower, D. R. (1983). Bedrock fracture parameters from the interpretation of well tides. *Journal of Geophysical Research: Solid Earth*, *88*(B6), 5025-5035.

Burbey, T. J., Hisz, D., Murdoch, L. C., & Zhang, M. (2012). Quantifying fractured crystalline-rock properties using well tests, earth tides and barometric effects. *Journal of Hydrology*, *414*, 317-328.

Rahi, K. A., & Halihan, T. (2013). Identifying aquifer type in fractured rock aquifers using harmonic analysis. *Groundwater*, *51*(1), 76-82.

Sedghi, M. M., & Zhan, H. (2016). Hydraulic response of an unconfined-fractured two-aquifer system driven by dual tidal or stream fluctuations. *Advances in water resources*, *97*, 266-278.

**Citation**: https://doi.org/10.5194/egusphere-2023-1727-RC1

---

## Author Comment (AC2)

**Reviewer 2**

In this Brief Note, the authors developed a more complete solution for the water level response to Earth tides in a leaky aquifer with aquitard storage and compressibility. The derivation of the solution is clearly given and easy to understand. On the other hand, there are some important points in the paper unclear to this referee, as detailed below:

First, it is unclear why the two Leaky & Storage models in Figure 2 are so very different. Both the amplitude ratio and phase difference for the model "Leaky & Storage (present study)" are functions of frequency at frequencies lower than that for the O1 tide, but both the amplitude ratio and phase difference become constant for the model "Leaky & Storage (present study') with K'=1e-14 m/s". Why are there such differences between the responses of the two models?

First of all, the "Leaky & Storage (present study') with K'=1e-14 m/s" was warried out with very low aquitard hydraulic conductivity so that the pressure and water transfer between aquitard and aquifer is unsignificant. Thus, it looks like the horizontal flux model and validate our model in such conditions.

For frequencies lower than $O_1$ using the parametrization of the study, amplitude is nearly equal to one, while phase shift is about zero for "Leaky & Storage (present study') with K'=1e-14 m/s". This constant behavior is the signature of the absence of well impact on groundwater level fluctuations and the absence of phase shift between the Earth tide strain and the aquifer level fluctuations. It means that the groundwater fluctuations of the aquifer are the same as the groundwater fluctuations in the well (absence of amplification/attenuation and phase shifts) and that there is no phase shift between the strain and the water pressure variations inside the aquifer.

For the "Leaky & Storage (present study)" model, the leaky conditions do provoke a phase shift and an amplitude modification as compared to purely confined conditions as observed in the new figure 4b. Such values of phase shift and amplitude modification do vary with the frequency because of the water pressure transfer between the aquitard and the aquifer.

[Figure]

*Figure 1: Example of the volumetric strain time series generated by the M2 Earth tide in a), which creates aquifer hydraulic head variations in b), resulting in well water level variations in c). The transmissivity (T) is $10^{-6}$ m²/s, storativity (S) is $7\,10^{-4}$, hydraulic conductivity of the aquitard (K') is $10^{-6}$ m/s, aquitard hydraulic diffusivity (D') is $10^{-4}$ m²/s, skin factor (sk) is -5 m, $z_i$ to -10 m, RKuB to 0.3, well casing radius (rc) and screen radius (rw) is 6.03 cm. B is set to 0.8 and $K_u$ to 10 GPa.*

Second, the captions of some diagrams are too brief that made the figures unnecessarily difficult to read. For example, the caption for Figure 5 states that the results are from 'using two aquifer models' without explaining which two leaky models. The caption also did not define the crosses or circles, which kept this referee guessing. Given the authors' comparison between their model and the model of Gao et al (2020), could the circles be that for the solution of Gao et al.?

We did a mistake in figure 5 caption, because only the present model developed in this study is presented. The red circles represent the best-fit. The comparison with the Gao et al solution is in appendix.

We corrected the captions in the new figure 6.

Finally, given the greater number of unknown parameters in the new solution, it is natural that the new solution may better reproduce the observed amplitude and frequency responses than previous models. The authors correctly pointed out that the solution is prone to non-uniqueness and *a priori* information is needed to reduce the number of unknowns in the solution.

Thanks for this relevant comment.